# Dynamic Event-Triggered Predictive Control for Interval Type-2 Fuzzy Systems with Imperfect Premise Matching

**DOI:** 10.3390/e23111452

**Published:** 2021-11-01

**Authors:** Jingfeng Zhou, Jianming Cao, Jing Chen, Aihua Hu, Jingxiang Zhang, Manfeng Hu

**Affiliations:** School of Science, Jiangnan University, Wuxi 214122, China; 6181202020@stu.jiangnan.edu.cn (J.Z.); 6181202015@stu.jiangnan.edu.cn (J.C.); 8201703038@jiangnan.edu.cn (J.C.); aihuahu@jiangnan.edu.cn (A.H.); zhangjingxiang@jiangnan.edu.cn (J.Z.)

**Keywords:** interval type-2 fuzzy systems, imperfect premise matching, dynamic event-triggered mechanism, predictive controller

## Abstract

This paper investigates the dynamic event-triggered predictive control problem of interval type-2 (IT2) fuzzy systems with imperfect premise matching. First, an IT2 fuzzy systems model is proposed, including a dynamic event-triggered mechanism, which can save limited network resources by reducing the number of data packets transmitted, and a predictive controller, which can predict the state of the system between the two successful transmitted instants to deal with unreliable communication networks. Then, according to the Lyapunov stability theory and imperfect premise matching method, sufficient conditions for system stabilization and the controller gain are obtained. Finally, the validity of the proposed method is demonstrated by the numerical examples.

## 1. Introduction

Networked control systems (NCSs) have attracted more attention during the past decades [1,2,3,4,5,6] due to their wide engineering applications, are control systems that connect various physical devices through a communication network with limited bandwidth in reality. To save communication resources and maintain system performance, the event-triggered mechanism (ETM) has been adopted recently to control the transmission of signals in the communication network. In [7,8,9,10,11], static ETM is used, in which the thresholds are always fixed scalars that do not really reflect the system dynamics, thereby leading to certain conservatism. It is desirable to have triggering laws whose threshold parameters are adaptively tuned depending on dynamical changes with the purpose of further reducing frequencies of signal transmissions. Following this line, a dynamic or adaptive ETM is designed in [12,13,14,15,16] by introducing an internal dynamic variable. However, some dynamic ETM might have a singular problem and degrade into a traditional time-triggered mechanism, which may restrict its use in practical applications. Recently, the multiplicative and additive internal dynamic variables of ETM are designed to avoid singular phenomena [17,18]. However, ETM may cause some practical problems due to the event triggered interval being too large for practical applications. Therefore, we set the maximum event-triggered interval to avoid these problems.

The static event-triggered predictive control method is proposed in [19,20], and its controller can obtain the estimated state of the system by introducing a predictor, which not only saves communication resources but also contributes to obtaining good system control performance. However, the static event-triggered predictive control method can not really reflect the system dynamics. Therefore, inspired by the aforementioned works, it is meaningful to design a dynamic event-triggered predictive control scheme.

The T-S fuzzy model plays an important role in the actual engineering, which can be used to represent some systems with nonlinear dynamics by the local linear subsystems under several IF-Then rules. In addition, it can also solve some nonlinear problems. For example, nonlinear disturbances can be represented by local linear disturbances. Therefore, the fuzzy model is widely used in practice and is very meaningful to study. Considering the nonuniform sampling, Wang et al. [21] proposed a fuzzy event-triggered asynchronous dissipation control method for the T-S fuzzy Markov jump system. Ma et al. [22] investigated the problem of adaptive fuzzy output feedback control for a class of stochastic nonlinear systems with full state constraints and actuator failures. Since the sensor and controller transmit signals through a communication network, it is not practical to assume that the fuzzy system and the fuzzy controller have the same premise variables. Therefore, an imperfect premise matching method is used to break this limitation. Asalm et al. [23] provided a fuzzy controller design method under ETM for a class of nonlinear systems with time-varying delays and mismatched premise variables. In reality, it is not easy to acquire membership functions because of the uncertainty of the parameters. To overcome this difficulty, the interval type-2 T-S (IT2) fuzzy model is proposed by bounding the membership function [24,25,26]. However, the problem of network packet loss in the communication network has not been solved in the above work, which inspired this work.

Motivated by the above discussions, the purpose of this paper is to design a dynamic event-triggered predictive controller for the IT2 fuzzy system, which has different premise variables from the IT2 fuzzy system and can compensate for the negative effects of the communication network. First of all, a new IT2 fuzzy system model is provided, which includes a dynamic ETM that can reduce the burden of communication networks and a predictive controller that can solve the problem of network packet loss. Then, the sufficient conditions for system stabilization are obtained by the Lyapunov stability theory and imperfect premise matching method, and the controller gain and event-triggered parameters are obtained by the given stabilization conditions. The main contributions are as follows:A novel IT2 fuzzy model is proposed, which unifies the dynamic ETM and the predictive control method in a framework to compensate the negative effect of network packet loss. Unlike the traditional T-S fuzzy model [27], it does not require the membership function be known by bounding it.A method of designing the dynamic event-triggered predictive controller containing global membership boundary information is provided to deal with imperfect premise mathing. Unlike the networked parallel distributed compensation method [28], it does not require the controller to have the same premise variables as the studied T-S fuzzy system by the imperfect premise matching method.

This paper is organized as follows. Section 2 is the system description including IT2 T-S fuzzy model and dynamic ETM. In Section 3, the stability of the system is analyzed, and sufficient conditions for system stabilization are obtained. Finally, a numerical example is given to illustrate the effectiveness of the design method.

*Notations:* Throughout this paper, the asterisk * in a matrix is used to denote a term that is induced by symmetry. X>0 (X≥0) means *X* is a symmetric and positive definite (positive semi-definite). *I* and 0 denote identity and zero matrix, respectively.

## 2. System Description

Figure 1 depicts the diagram of IT2 fuzzy systems with dynamical fuzzy event-triggered predictive controller (FETPC). The event-triggered predictive control method in this paper includes ETM1 to transmit sampled data to the predictor, ETM2 to transmit the predicted state to the controller, and the networked data-dropout compensator (NDC) to store the predicted data packets. The sensor and FETPC are connected through a communication network, while the FETPC and actuator are directly connected without communication networks. To make this framework more clear, detailed descriptions of some components will be given below.

### 2.1. IT2 Fuzzy Model

Considering the following networked IT2 fuzzy systems.

Rule ϵ: IF $f_{1}\left(x\left(k\right)\right)$ is $M_{1}^{\epsilon }$, and ⋯, and $f_{p}\left(x\left(k\right)\right)$ is $M_{p}^{\epsilon }$, THEN:(1)$$\begin{array}{c} x\left(k+1\right)=A_{\epsilon }x\left(k\right)+B_{\epsilon }u\left(k\right), \end{array}$$
where $f_{\nu }\left(x\left(k\right)\right)$ and $M_{v}^{\epsilon }$ (ν=1,2,⋯,p; ϵ=1,2,⋯,r) denote the premise variables and the fuzzy sets, $x\left(k\right)\in R^{n}$ and $u\left(k\right)\in R^{n}$ are system state and control input, respectively. $A_{\epsilon }$ and $B_{\epsilon }$ are constant matrices with appropriate dimensions. The activation intensity of rule ϵ can be defined:$$W_{\epsilon }\left(x\left(k\right)\right)=\left[{\underline{h}}_{\epsilon }\left(x\left(k\right)\right)\;{\bar{h}}_{\epsilon }\left(x\left(k\right)\right)\right],$$
where ${\underline{h}}_{\epsilon }\left(x\left(k\right)\right)=\prod_{\nu =1}^{p}{\underline{\mu }}_{M_{\nu }^{\epsilon }}\left(f_{\nu }\left(x\left(k\right)\right)\right)$ and ${\bar{h}}_{\epsilon }\left(x\left(k\right)\right)=\prod_{\nu =1}^{p}{\bar{\mu }}_{M_{\nu }^{\epsilon }}\left(f_{\nu }\left(x\left(k\right)\right)\right)$ with ${\underline{\mu }}_{M_{\nu }^{\epsilon }}\left(f_{\nu }\left(x\left(k\right)\right)\right)\in \left[0,1\right]$ and ${\bar{\mu }}_{M_{\nu }^{\epsilon }}\left(f_{\nu }\left(x\left(k\right)\right)\right)\in \left[0,1\right]$ denoting the lower and upper grades of membership, respectively.

Then the system (Equation 1) can be formulated by:(2)$$\begin{array}{c} x\left(k+1\right)=\sum_{\epsilon =1}^{r}h_{\epsilon }\left(x\left(k\right)\right)\left[A_{\epsilon }x\left(k\right)+B_{\epsilon }u\left(k\right)\right], \end{array}$$
in which $h_{\epsilon }\left(x\left(k\right)\right)={\underline{\kappa }}_{\epsilon }{\underline{h}}_{\epsilon }\left(x\left(k\right)\right)+{\bar{\kappa }}_{\epsilon }{\bar{h}}_{\epsilon }\left(x\left(k\right)\right)$ satisfies $0\leq h_{\epsilon }\left(x\left(k\right)\right)\leq 1$ and $\sum_{\epsilon =1}^{r}h_{\epsilon }\left(x\left(k\right)\right)=1$. ${\underline{\kappa }}_{\epsilon }\in \left[0,1\right]$ and ${\bar{\kappa }}_{\epsilon }\in \left[0,1\right]$ are the nonlinear weighting functions that satisfy ${\underline{\kappa }}_{\epsilon }+{\bar{\kappa }}_{\epsilon }=1$.

**Remark** **1.**
*Unlike the T-S fuzzy model [19] for predictive control of networked nonlinear system with imperfect premise matching, the membership of IT2 fuzzy model is no longer a definite value, but in an interval. IT2 fuzzy model not only extends the traditional T-S fuzzy model, but also has the characteristic of dealing with uncertainty.*


### 2.2. Dynamic ETM

In order to save limited communication resources, a dynamic ETM is designed to release the sampled signals to the communication network. Let $e\left(k\right)=x\left(k\right)-x\left(k_{n}\right)$ be the error between the current state x(k) and the latest triggered state $x(k_{n})$. Then, the next event-triggered instant depends on the ETM1 (see Figure 1).
(3)$$\begin{array}{c} \begin{array}{cc} k_{n+1}= & min\{k_{n}+T,T_{k}\},\\ T_{k}= & min\{k|k>k_{n},\frac{1}{\rho }\phi \left(k\right)+\delta x^{T}\left(k\right)x\left(k\right)\\ & -e^{T}\left(k\right)e\left(k\right)\leq 0\},n=0,1,2,\cdots , \end{array} \end{array}$$
where 0<δ<1 and ρ>0 are given constants, *T* is the upper bound of the interval of adjacent triggering instants. The variable ϕ(k) is designed as:(4)$$\begin{array}{c} \phi \left(k+1\right)=\tau \phi \left(k\right)+\delta x^{T}\left(k\right)x\left(k\right)-e^{T}\left(k\right)e\left(k\right), \end{array}$$
where τ∈(0,1) is a given constant and $\phi \left(0\right)=\phi_{0}>0$. Parameters τ and ρ satisfied τρ>1. In interval $\left[k_{n},k_{n+1}\right)$, ϕ(k)≥0 can be obtained by combining (Equation 3), (Equation 4) and τρ>1.

**Remark** **2.**
*The dynamic ETM1 contains an internal dynamic variable ϕ(k), which can dynamically adjust the intensity of the ETM1 according to the system state. It can be seen that when ϕ(k)→0, the dynamic ETM1 becomes the static ETM [19].*


### 2.3. FETPC under Premise Matching

Due to the limited network resources and the unreliability of the communication network, some sampled data will not be transmitted to the controller, so the predictor is set in the controller. Considering the controller model can not share the premise variables with the system, the model of fuzzy predictive controller is described as

Rule *w*: IF $g_{1}\left(\hat{x}\left(k\right)\right)$ is $N_{1}^{w}$ and ⋯ and $g_{q}\left(\hat{x}\left(k\right)\right)$ is $N_{q}^{w}$, then,
(5)$$\begin{array}{c} \hat{x}\left(k+1\right)={\hat{A}}_{w}\hat{x}\left(k\right)+{\hat{B}}_{w}u\left(k\right), \end{array}$$

Rule *l*: IF $g_{1}\left(\hat{x}\left(k\right)\right)$ is $N_{1}^{l}$ and ⋯ and $g_{q}\left(\hat{x}\left(k\right)\right)$ is $N_{q}^{l}$, then,
(6)$$\begin{array}{c} \hat{u}\left(k\right)=K_{l}\hat{x}\left(k\right), \end{array}$$
where $g_{\gamma }\left(\hat{x}\left(k\right)\right)$ and $N_{\gamma }^{w,l}$ (w,l=1,2,⋯,o;γ=1,2,⋯,q) denote the premise variables and the fuzzy sets, $\hat{x}\left(k\right)$ is the predicted system state, ${\hat{A}}_{w}$ and ${\hat{B}}_{w}$ represent constant matrices with appropriate dimensions. Similarly, the model of fuzzy predictive controller can be described as:(7)$$\begin{array}{c} \hat{x}\left(k+1\right)=\sum_{w=1}^{o}\eta_{w}\left(g\left(\hat{x}\left(k\right)\right)\right)\left[{\hat{A}}_{w}\hat{x}\left(k\right)+{\hat{B}}_{w}u\left(k\right)\right], \end{array}$$
(8)$$\begin{array}{c} \hat{u}\left(k\right)=\sum_{l=1}^{o}\eta_{l}\left(g\left(\hat{x}\left(k\right)\right)\right)K_{l}\hat{x}\left(k\right), \end{array}$$
where:$$\begin{array}{c} \eta_{w}={\underline{\kappa }}_{w}{\underline{W}}_{w}\left(\hat{x}\left(k\right)\right)+{\bar{\kappa }}_{w}{\bar{W}}_{w}\left(\hat{x}\left(k\right)\right),0\leq \eta_{w}\left(\hat{x}\left(k\right)\right)\leq 1,\\ \sum_{w=1}^{o}\eta_{w}\left(\hat{x}\left(k\right)\right)=1,{\underline{\kappa }}_{w}\in \left[0\;1\right],{\bar{\kappa }}_{w}\in \left[0\;1\right],{\underline{\kappa }}_{w}+{\bar{\kappa }}_{w}=1,\\ \eta_{l}={\underline{\kappa }}_{l}{\underline{W}}_{l}\left(\hat{x}\left(k\right)\right)+{\bar{\kappa }}_{l}{\bar{W}}_{l}\left(\hat{x}\left(k\right)\right),0\leq \eta_{l}\left(\hat{x}\left(k\right)\right)\leq 1,\\ \sum_{w=1}^{o}\eta_{l}\left(\hat{x}\left(k\right)\right)=1,{\underline{\kappa }}_{l}\in \left[0\;1\right],{\bar{\kappa }}_{l}\in \left[0\;1\right],{\underline{\kappa }}_{l}+{\bar{\kappa }}_{l}=1. \end{array}$$

Noting that the premise variables of the predictor and the controller are the same, but they are inconsistent with the premise variables of the fuzzy system.

### 2.4. Model of Networked T-S Fuzzy Systems

In this section, we will carefully analyze the process of data transmission in the network and design the program for the event-triggered predictor.

Denote $t_{s_{i}}\left(i=1,2,...\right)$ the time series when trigger states are successfully sent to the controller by ETM1 (Equation 3), the closed-loop system (Equation 1) can be predicted as:(9)$$\begin{array}{c} \begin{array}{cc} \hat{x}\left({\hat{t}}_{s_{i}+j}+k+1|t_{s_{i}}\right) & =\sum_{w=1}^{o}\eta_{w}g\left(\hat{x}\right)[{\hat{A}}_{w}\hat{x}\left({\hat{t}}_{s_{i}+j}+k|t_{s_{i}}\right)\\ \; & +{\hat{B}}_{w}\sum_{l=1}^{o}\eta_{l}g\left(\hat{x}\right)K_{l}\hat{x}\left({\hat{t}}_{s_{i}}+j|t_{s_{i}}\right), \end{array} \end{array}$$
where $k\in \{0,1,2,\cdots ,{\hat{t}}_{s_{i}+j+1}-{\hat{t}}_{s_{i}+j}-1\}$ and $j\in \{0,1,2,\cdots ,\theta_{i}\}$. In the time interval [$t_{s_{i}}$, $t_{s_{i+1}}$), the states of the system (Equation 9) are $x\left(t_{s_{i}}\right),\;\hat{x}\left(t_{s_{i}}+1\right),\;\hat{x}\left(t_{s_{i}}+2\right),\;\cdots ,\;\hat{x}\left({\hat{t}}_{s_{i}+1}\right),\;\hat{x}\left({\hat{t}}_{s_{i}+1}+1\right),\;\hat{x}\left({\hat{t}}_{s_{i}+1}+2\right),\;\cdots ,\;\hat{x}\left({\hat{t}}_{s_{i}+\theta_{i}-1}\right),\;\hat{x}\left({\hat{t}}_{s_{i}+\theta_{i}-1}+1\right),\;\hat{x}\left({\hat{t}}_{s_{i}+\theta_{i}-1}+2\right),\;\cdots ,\;\hat{x}\left({\hat{t}}_{s_{i}+\theta_{i}}\right)$. Note that when j=0, we get ${\hat{t}}_{s_{i}+j}=t_{s_{i}}$ and $\hat{x}\left({\hat{t}}_{s_{i}}|t_{s_{i}}\right)=x\left(t_{s_{i}}\right)$.

In the communication network, there is the phenomenon of network packet loss, in the following, we will make an assumption on the packet loss.

**Assumption** **1.**
*The upper bounded of the number of consecutive loss-data occurring is σ. When the data packet is lost, the triggered state will not be received by the controller, and the controller continues to use the predicted state.*


In (Equation 9), the predictive event-triggered instants ${\hat{t}}_{s_{i}+j+1}$ are determined by ETM2 (see Figure 1),
(10)$$\begin{array}{c} \begin{array}{cc} {\hat{t}}_{s_{i}+j+1}= & min\{{\hat{t}}_{s_{i}+j}+T,T_{s_{i}+j}\},\\ T_{s_{i}+j}= & min\{k|k>{\hat{t}}_{s_{i}+j},\frac{1}{\rho }\hat{\phi }+\delta {\hat{x}}^{T}\left(t|t_{s_{i}}\right)\hat{x}\left(t|t_{s_{i}}\right)\\ & -{\hat{e}}^{T}\left(t\right)\hat{e}\left(t\right)\leq 0\},j=0,1,2,\cdots ,\theta_{i}, \end{array} \end{array}$$
and the variable $\hat{\phi }\left(k\right)$ is designed as:(11)$$\begin{array}{c} \hat{\phi }\left(k+1\right)=\tau \hat{\phi }\left(k\right)+\delta {\hat{x}}^{T}\left(k\right)\hat{x}\left(k\right)-{\hat{e}}^{T}\left(k\right)\hat{e}\left(k\right), \end{array}$$
where ρ, δ, τ and *T* are same as in (Equation 3) and (Equation 4). Obviously, $\hat{\phi }\left(k\right)\geq 0$ can be obtained when $\hat{\phi }\left(0\right)={\hat{\phi }}_{0}\geq 0$. Defining $\hat{e}\left(t\right)=\hat{x}\left(t|t_{s_{i}}\right)-\hat{x}\left({\hat{t}}_{s_{i}+j}|t_{s_{i}}\right)$. The predictive event-triggered instants are ${\left\{{\hat{t}}_{s_{i}+j}\right\}}_{j=1}^{\theta_{i}}$ and it satisfies ${\hat{t}}_{s_{i}+\theta_{i}}\leq t_{s_{i}}+T\mp \sigma <{\hat{t}}_{s_{i}+\theta_{i}+1}$ from the Assumption 1.

The predictive control signals can be represented as:(12)$$\begin{array}{c} \begin{array}{c} \hat{u}\left({\hat{t}}_{s_{i}+j}|t_{s_{i}}\right)=\sum_{l=1}^{o}\eta_{l}\left(g\left(\hat{x}\right)\right)K_{l}\hat{x}\left({\hat{t}}_{s_{i}+j}|t_{s_{i}}\right),j\in \left\{0,1,\cdots ,\theta_{i}\right\} \end{array} \end{array}$$
and the control sequence in (Equation 12) is expressed as:(13)$$\begin{array}{c} \begin{array}{c} U_{t_{s_{i}}}=\left[u\left(t_{s_{i}}\right),\hat{u}\left({\hat{t}}_{s_{i}+1}|t_{s_{i}}\right),\cdots ,\hat{u}\left({\hat{t}}_{s_{i}+\theta_{i}}|t_{s_{i}}\right)\right] \end{array} \end{array}$$

**Remark** **3.**
*If ETM2 (Equation 10) is not introduced, then the control sequence will be expressed as:*

$$\begin{array}{cc} {\tilde{U}}_{t_{s_{i}}}=[ & u\left(t_{s_{i}}\right),\hat{u}\left(t_{s_{i}}+1\right),\cdots ,\\ & \hat{u}\left({\hat{t}}_{s_{i}+1}|t_{s_{i}}\right),\hat{u}\left({\hat{t}}_{s_{i}+1}+1|t_{s_{i}}\right),\cdots ,\\ & \hat{u}\left({\hat{t}}_{s_{i}+2}|t_{s_{i}}\right),\hat{u}\left({\hat{t}}_{s_{i}+2}+1|t_{s_{i}}\right),\cdots ,\\ & \hat{u}\left({\hat{t}}_{s_{i}+\theta_{i}}|t_{s_{i}}\right)]. \end{array}$$


*Compared with $U_{t_{s_{i}}}$, the complexity and size of ${\tilde{U}}_{t_{s_{i}}}$ are larger. Therefore, ETM2 (Equation 10) can save computing and storage resources.*


Until now, (Equation 2) and (Equation 7) can be expressed as the closed loop system:(14)$$\begin{array}{c} x\left(t+1\right)=\sum_{\epsilon =1}^{r}h_{\epsilon }\left(x\right)\left[A_{\epsilon }x\left(t\right)+B_{\epsilon }\hat{u}\left({\hat{k}}_{s_{i}+j}|t_{s_{i}}\right)\right], \end{array}$$
(15)$$\begin{array}{c} \hat{x}\left(t+1\right)=\sum_{w=1}^{o}\eta_{w}\left(\hat{x}\right)\left[{\hat{A}}_{w}\hat{x}\left(t\right)+{\hat{B}}_{w}\hat{u}\left({\hat{t}}_{s_{i}+j}|t_{s_{i}}\right)\right],t\in \Phi_{mj}^{i}, \end{array}$$
where $\Phi_{mj}^{i}≜\left[t_{s_{i}},t_{s_{i+1}}\right)\cap \left[t_{s_{i}+m},t_{s_{i}+m+1}\right)\cap \left[{\hat{t}}_{s_{i}+j},{\hat{t}}_{s_{i}+j+1}\right)$, and $\left[t_{s_{i}},t_{s_{i}+1}\right)=\cup_{m=0}^{s_{i+1}-s_{i}-1}\cup_{j=0}^{\theta_{i}}\Phi_{mj}^{i}$. By defining ${\hat{e}}_{ij}\left(t\right)=\hat{x}\left(t\right)-\hat{x}\left({\hat{t}}_{s_{i}+j}|t_{s_{i}}\right),e_{im}=x\left(t\right)-x\left(t_{s_{i}+m}\right)$, $\beta \left(t\right)=x\left(t\right)-\hat{x}\left(t\right)$ and $\alpha ={\left[x^{T}\left(t\right),\beta^{T}\left(t\right)\right]}^{T}$, the system (Equation 14) and (Equation 15) can be uniformly expressed as:(16)$$\begin{array}{c} \alpha \left(t+1\right)=\sum_{\epsilon =1}^{o}\sum_{w=1}^{o}\sum_{l=1}^{o}h_{\epsilon }\left(x\right)\eta_{w}\left(\hat{x}\right)\eta_{l}\left(\hat{x}\right)\left[\chi_{\epsilon wl}\right], \end{array}$$
where:$$\begin{array}{c} \chi_{\epsilon wl}=\Pi_{\epsilon wl}\alpha \left(t\right)+\Xi_{\epsilon wl}{\hat{e}}_{ij}\left(t\right),\;\Xi_{\epsilon wl}=\left[\begin{array}{c} -B_{\epsilon }K_{l}\\ -\left(B_{\epsilon }-{\hat{B}}_{\epsilon }\right)K_{l} \end{array}\right], \end{array}$$
$$\begin{array}{c} \chi_{\epsilon wl}=\left[\begin{array}{cc} A_{\epsilon }+B_{\epsilon }K_{l} & -B_{\epsilon }K_{l}\\ A_{\epsilon }-{\hat{A}}_{w}+\left(B_{\epsilon }-{\hat{B}}_{w}\right)K_{l} & {\hat{A}}_{\epsilon }-\left(B_{\epsilon }-{\hat{B}}_{w}\right)K_{l} \end{array}\right]. \end{array}$$

**Remark** **4.**
*In the IT2 fuzzy system model, the premise variables of the system (Equation 2) and FETPC (Equation 8) are imperfectly matched, which is expressed as $h_{\epsilon }\neq \eta_{l}$ in (Equation 16). Perfect premise matching can be regarded as a special case of this paper, which means that the design method of this paper can be used in any situations regardless of imperfect/perfect matching.*


Before presenting the main results, we also need the following Lemma.

**Lemma** **1.**
*[29] Given matrices $Q_{i}\left(i=1,\cdots ,s\right)$ and positive semi-definite matrix P, if $\sum_{i=1}^{s}v_{i}=1$ and $0\leq v_{i}\leq 1$ exist, the following inequality holds,*

$$(\sum_{i=1}^{s}v_{i}Q_{i}{)}^{T}P(\sum_{i=1}^{s}v_{i}Q_{i})\leq \sum_{i=1}^{s}v_{i}Q_{i}^{T}PQ_{i}.$$



## 3. Main Results

In this section, we analyze the asymptotic stability of the system (Equation 16) under the dynamic ETM1 (Equation 3) and ETM2 (Equation 10), and the stability criteria are established.

**Theorem** **1.**
*Given parameters ρ>0,δ>0,0<τ<1, matrices $K_{l}$ and membership function satisfying $\eta_{l}\left(\hat{x}\right)-\gamma_{l}h_{l}\left(x\right)\geq 0\left(0<\gamma_{l}<1\right)$, the closed-loop system (Equation 16) can achieve asymptotically stable under (Equation 3) and (Equation 10), if there exist matrices P>0 and arbitrary matrices $\Lambda_{\epsilon },\Lambda_{l}$ with appropriate dimensions for ϵ,w,l=1,2,⋯,o satisfying:*

(17)
$$\begin{array}{c} \Psi_{\epsilon wl}-\Lambda_{l}<0, \end{array}$$


(18)
$$\begin{array}{c} \gamma_{\epsilon }\Psi_{\epsilon w\epsilon }-\gamma_{\epsilon }\Lambda_{\epsilon }+\Lambda_{\epsilon }<0, \end{array}$$


(19)
$$\begin{array}{c} \gamma_{l}\Psi_{\epsilon wl}+\gamma_{\epsilon }\Psi_{lw\epsilon }-\gamma_{l}\Lambda_{\epsilon }-\gamma_{\epsilon }\Lambda_{l}+\Lambda_{\epsilon }+\Lambda_{l}<0, \end{array}$$


$$\Psi_{\epsilon wl}≜\left[\begin{array}{cccccc} -P+\Omega & * & * & * & * & *\\ 0 & -\upsilon_{1}I & * & * & * & *\\ 0 & 0 & -\upsilon_{1}I & * & * & *\\ 0 & 0 & 0 & \upsilon_{2}I & * & *\\ 0 & 0 & 0 & 0 & \upsilon_{2}I & *\\ P\Pi_{\epsilon wl} & P\Xi_{\epsilon wl} & 0 & 0 & 0 & -P \end{array}\right],$$


$$\upsilon_{1}=\left(\frac{1}{\rho }+\delta \right),\;\upsilon_{2}=\frac{\tau -1+c}{\rho },\;\upsilon_{3}=\frac{\delta }{\rho }+c\delta ,\;\Omega ≜\left[\begin{array}{cc} 2\upsilon_{3}I & -\upsilon_{3}I\\ -\upsilon_{3}I & \upsilon_{3}I \end{array}\right].$$



**Proof.** Choose a Lyapunov function as:
(20)$$\begin{array}{c} V\left(\alpha \left(k\right),\phi \left(k\right)\right)=\alpha^{T}\left(k\right)P\alpha \left(k\right)+\frac{1}{\rho }\phi \left(k\right)+\frac{1}{\rho }\hat{\phi }\left(k\right), \end{array}$$
then,
(21)$$\begin{array}{c} \begin{array}{cc} \Delta V(\alpha (k),\phi (k))= & \alpha^{T}\left(k+1\right)P\alpha \left(k+1\right)-\alpha^{T}\left(k\right)P\alpha \left(k\right)\\ \; & +\frac{1}{\rho }\phi \left(k+1\right)-\frac{1}{\rho }\phi \left(k\right)+\frac{1}{\rho }\hat{\phi }\left(k+1\right)-\frac{1}{\rho }\hat{\phi }\left(k\right). \end{array} \end{array}$$Through the dynamic ETM (Equation 3), for any $t\in (t_{s_{i}+m},t_{s_{i}+m+1})$,
(22)$$\begin{array}{c} \frac{1}{\rho }\phi \left(k\right)+\delta x^{T}\left(k\right)x\left(k\right)-e^{T}\left(k\right)e\left(k\right)\geq 0, \end{array}$$
which implies that for any c>0,
(23)$$\begin{array}{c} \begin{array}{cc} \; & \frac{1}{\rho }\phi \left(k+1\right)-\frac{1}{\rho }\phi \left(k\right)\\ & \leq \frac{1}{\rho }\phi \left(k+1\right)-\frac{1}{\rho }\phi \left(k\right)+c\left(\frac{1}{\rho }\phi \left(k\right)+\delta x^{T}\left(k\right)x\left(k\right)-e^{T}\left(k\right)e\left(k\right)\right)\\ \; & =\frac{\tau -1+c}{\rho }\phi \left(k\right)+\left(\frac{\delta }{\rho }+c\delta \right)x^{T}\left(k\right)x\left(k\right)-\left(\frac{1}{\rho }+\delta \right)e^{T}\left(k\right)e\left(k\right)). \end{array} \end{array}$$Similarly, from the dynamic ETM (Equation 10), for any $t\in ({\hat{t}}_{s_{i}+j},{\hat{t}}_{s_{i}+j+1})$ and c>0,
(24)$$\begin{array}{cc} & \frac{1}{\rho }\hat{\phi }\left(k+1\right)-\frac{1}{\rho }\hat{\phi }\left(k\right)\\ & \leq \frac{1}{\rho }\hat{\phi }\left(k+1\right)-\frac{1}{\rho }\hat{\phi }\left(k\right)\\ & +c(\frac{1}{\rho }\hat{\phi }\left(k\right)+\delta \hat{x}{\left(k\right)}^{T}\hat{x}\left(k\right)-{\hat{e}}_{ij}^{T}\left(t\right){\hat{e}}_{ij}\left(k\right))\\ & =\frac{\tau -1+c}{\rho }\hat{\phi }\left(k\right)+\left(\frac{\delta }{\rho }+c\delta \right)\hat{x}{\left(k\right)}^{T}\hat{x}\left(k\right)\\ \; & -\left(\frac{1}{\rho }+\delta \right){\hat{e}}_{ij}^{T}\left(k\right){\hat{e}}_{ij}\left(k\right)). \end{array}$$On the other hand, by using Lemma 1, one has:
(25)$$\begin{array}{cc} \alpha^{T}\left(k+1\right)P\alpha \left(k+1\right)= & \{\sum_{\epsilon =1}^{o}\sum_{w=1}^{o}\sum_{l=1}^{o}h_{\epsilon }\left(x\right)\eta_{w}\left(\hat{x}\right)\eta_{l}\left(\hat{x}\right)\left[\Pi_{\epsilon wl}\alpha \left(k\right)+\Xi_{\epsilon wl}{\hat{e}}_{ij}\left(k\right)\right]{\}}^{T}\\ & P\{\sum_{\epsilon =1}^{o}\sum_{w=1}^{o}\sum_{l=1}^{o}h_{\epsilon }\left(x\right)\eta_{w}\left(\hat{x}\right)\eta_{l}\left(\hat{x}\right)\left[\Pi_{\epsilon wl}\alpha \left(k\right)+\Xi_{\epsilon wl}{\hat{e}}_{ij}\left(k\right)\right]\}\\ \leq & \sum_{\epsilon =1}^{o}h_{\epsilon }\left(x\right)\{\sum_{w=1}^{o}\sum_{l=1}^{o}\eta_{w}\left(\hat{x}\right)\eta_{l}\left(\hat{x}\right)\left[\Pi_{\epsilon wl}\alpha \left(k\right)+\Xi_{\epsilon wl}{\hat{e}}_{ij}\left(k\right)\right]{\}}^{T}\\ & P\{\sum_{w=1}^{o}\sum_{l=1}^{o}\eta_{w}\left(\hat{x}\right)\eta_{l}\left(\hat{x}\right)\left[\Pi_{\epsilon wl}\alpha \left(k\right)+\Xi_{\epsilon wl}{\hat{e}}_{ij}\left(k\right)\right]\}\\ \leq & \sum_{\epsilon =1}^{o}\sum_{w=1}^{o}h_{\epsilon }\left(x\right)\eta_{w}\left(\hat{x}\right)\{\sum_{l=1}^{o}\eta_{l}\left(\hat{x}\right)\left[\Pi_{\epsilon wl}\alpha \left(k\right)+\Xi_{\epsilon wl}{\hat{e}}_{ij}\left(k\right)\right]{\}}^{T}\\ & P\{\sum_{l=1}^{o}\eta_{l}\left(\hat{x}\right)\left[\Pi_{\epsilon wl}\alpha \left(k\right)+\Xi_{\epsilon wl}{\hat{e}}_{ij}\left(k\right)\right]\}\\ \leq & \sum_{\epsilon =1}^{o}\sum_{w=1}^{o}\sum_{l=1}^{o}h_{\epsilon }\left(x\right)\eta_{w}\left(\hat{x}\right)\eta_{l}\left(\hat{x}\right)[\Pi_{\epsilon wl}\alpha \left(k\right)+\Xi_{\epsilon wl}{\hat{e}}_{ij}\left(k\right){]}^{T}\\ \; & P[\Pi_{\epsilon wl}\alpha \left(k\right)+\Xi_{\epsilon wl}{\hat{e}}_{ij}\left(k\right)]. \end{array}$$By considering (Equation 21)–(Equation 25) and Schur complement, we obtain:
(26)$$\begin{array}{cc} \Delta V(\alpha (k),\phi (k)) & \leq \sum_{\epsilon =1}^{o}\sum_{w=1}^{o}\sum_{l=1}^{o}h_{\epsilon }\left(x\right)\eta_{w}\left(\hat{x}\right)\eta_{l}\left(\hat{x}\right)\{[\Pi_{\epsilon wl}\alpha \left(k\right)+\Xi_{\epsilon wl}{\hat{e}}_{ij}\left(k\right){]}^{T}\\ & P[\Pi_{\epsilon wl}\alpha \left(k\right)+\Xi_{\epsilon wl}{\hat{e}}_{ij}\left(k\right)]+\frac{\tau -1+c}{\rho }\phi \left(k\right)\\ & -\left(\frac{1}{\rho }+\delta \right)e^{T}\left(k\right)e\left(k\right))+\frac{\tau -1+c}{\rho }\hat{\phi }\left(k\right)\\ & -\left(\frac{1}{\rho }+\delta \right){\hat{e}}_{ij}^{T}\left(k\right){\hat{e}}_{ij}\left(k\right))+\alpha^{T}\left(k\right)\Omega \alpha \left(k\right)-\alpha^{T}\left(k\right)P\alpha \left(k\right)\}\\ \; & \leq \sum_{\epsilon =1}^{o}\sum_{w=1}^{o}\sum_{l=1}^{o}h_{\epsilon }\left(x\right)\eta_{w}\left(\hat{x}\right)\eta_{l}\left(\hat{x}\right)\xi^{T}\Psi_{\epsilon wl}\xi <0, \end{array}$$
where $\xi^{T}≜\left[\alpha^{T}\left(t\right),e,{\hat{e}}_{ij}^{T},x\left(t\right),{\sqrt{\hat{\phi }\left(t\right)}}^{T},{\sqrt{\phi (t)}}^{T}\right]$, and we get:
(27)$$\begin{array}{cc} & \sum_{\epsilon =1}^{o}\sum_{w=1}^{o}\sum_{l=1}^{o}h_{\epsilon }\left(x\right)\eta_{w}\left(\hat{x}\right)\left[h_{l}\left(x\right)-\eta_{l}\left(\hat{x}\right)\right]\Lambda_{\epsilon }\\ & =\sum_{\epsilon =1}^{o}\sum_{w=1}^{o}h_{\epsilon }\left(x\right)\eta_{w}\left(\hat{x}\right)[\sum_{l=1}^{o}h_{l}\left(x\right)-\sum_{l=1}^{o}\eta_{l}]\Lambda_{\epsilon }\\ \; & =\sum_{\epsilon =1}^{o}\sum_{w=1}^{o}h_{\epsilon }\left(x\right)\eta_{w}\left(\hat{x}\right)\left(1-1\right)\Lambda_{\epsilon }=0, \end{array}$$
where $\Lambda_{\epsilon }$ is arbitrary matrix. Combining (Equation 26) and (Equation 27), we get:
(28)$$\begin{array}{cc} & \sum_{\epsilon =1}^{o}\sum_{w=1}^{o}\sum_{l=1}^{o}h_{\epsilon }\left(x\right)\eta_{w}\left(\hat{x}\right)\eta_{l}\left(\hat{x}\right)\Psi_{\epsilon wl}\\ & \leq \sum_{\epsilon =1}^{o}\sum_{w=1}^{o}h_{\epsilon }^{2}\left(x\right)\eta_{w}\left(\hat{x}\right)\left(\gamma_{\epsilon }\Psi_{\epsilon w\epsilon }-\gamma_{\epsilon }\Lambda_{\epsilon }+\Lambda_{\epsilon }\right)\\ \; & +\sum_{\epsilon =1}^{o}\sum_{w=1}^{o}\sum_{l=1}^{o}h_{\epsilon }\left(x\right)\eta_{w}\left(\hat{x}\right)\left(\eta_{l}\left(\hat{x}\right)-\gamma_{l}h_{l}\left(x\right)\right)\left(\Psi_{\epsilon wl}-\Lambda_{l}\right)\\ & +\sum_{\epsilon =1}^{o}\sum_{w=1}^{o}\sum_{l<\epsilon }^{o}h_{\epsilon }\left(x\right)\eta_{w}\left(\hat{x}\right)h_{l}\left(x\right)\left(\gamma_{l}\Psi_{\epsilon wl}+\gamma_{\epsilon }\Psi_{lw\epsilon }-\gamma_{l}\Lambda_{\epsilon }-\gamma_{\epsilon }\Lambda_{l}+\Lambda_{\epsilon }+\Lambda_{l}\right), \end{array}$$
where $\eta_{l}\left(\hat{x}\right)-\gamma_{l}h_{l}\left(x\right)\geq 0$ for all *l*. Let (Equation 17), (Equation 18) and (Equation 19) hold for all ϵ,w,l=1,2,⋯,o, then the following can be obtained:
(29)$$\begin{array}{c} \Delta V(\alpha (t),\phi (t))<0. \end{array}$$Obviously, there is a scalar ι>0 satisfying $\Delta V\left(\alpha \left(t\right),\phi \left(t\right)\right)\leq -\iota ∥\xi^{2}∥$ for all ξ≠0. Therefore, the system (Equation 16) achieves asymptotical stablility. □

Although Theorem 1 has guaranteed the stability of the closed-loop system (Equation 16), in order to find the parameters of FETPC and ETM, Theorem 2 is given.

**Theorem** **2.**
*Given parameters ρ>0,δ>0,0<τ<1, matrices $K_{l}$ and membership function satisfying $\eta_{l}\left(\hat{x}\right)-\gamma_{l}h_{l}\left(x\right)\geq 0\left(0<\gamma_{l}<1\right)$, the closed-loop system (Equation 16) can achieve asymptotically stable under (Equation 3) and (Equation 10), if there exist matrices $\bar{\Upsilon }>0$ and arbitrary matrices ${\bar{\Lambda }}_{\epsilon },{\bar{\Lambda }}_{l}$ with appropriate dimensions for ϵ,w,l=1,2,⋯,o satisfying:*

(30)
$$\begin{array}{c} {\bar{\Psi }}_{\epsilon wl}-{\bar{\Lambda }}_{l}\leq 0, \end{array}$$


(31)
$$\begin{array}{c} \gamma_{\epsilon }{\bar{\Psi }}_{\epsilon w\epsilon }-\gamma_{\epsilon }{\bar{\Lambda }}_{\epsilon }+{\bar{\Lambda }}_{\epsilon }\leq 0, \end{array}$$


(32)
$$\begin{array}{c} \gamma_{l}{\bar{\Psi }}_{\epsilon wl}+\gamma_{\epsilon }{\bar{\Psi }}_{lw\epsilon }-\gamma_{l}{\bar{\Lambda }}_{\epsilon }-\gamma_{\epsilon }{\bar{\Lambda }}_{l}+{\bar{\Lambda }}_{\epsilon }+{\bar{\Lambda }}_{l}\leq 0, \end{array}$$


*where,*

$${\bar{\Psi }}_{\epsilon wl}≜\left[\begin{array}{ccccccc} {\bar{\Psi }}_{11} & {\bar{\Psi }}_{12} & * & * & * & * & *\\ 0 & 0 & {\bar{\Psi }}_{23} & * & * & * & *\\ 0 & 0 & 0 & {\bar{\Psi }}_{34} & * & * & *\\ 0 & 0 & 0 & 0 & {\bar{\Psi }}_{45} & * & *\\ 0 & 0 & 0 & 0 & 0 & {\bar{\Psi }}_{56} & *\\ {\bar{\Psi }}_{61} & {\bar{\Psi }}_{62} & {\bar{\Psi }}_{63} & 0 & 0 & 0 & -P \end{array}\right],$$


*with,*

$${\bar{\Psi }}_{11}=\left[\begin{array}{c} 2{\bar{\upsilon }}_{3}I-\bar{\Upsilon }\\ -{\bar{\upsilon }}_{3}I \end{array}\right],{\bar{\Psi }}_{12}=\left[\begin{array}{c} {}*\\ 2{\bar{\upsilon }}_{3}I-\bar{\Upsilon } \end{array}\right],{\bar{\Psi }}_{61}=\left[\begin{array}{c} A_{\epsilon }\bar{\Upsilon }+B_{\epsilon }Y_{l}\\ \left(A_{\epsilon }-{\hat{A}}_{w}\right)\bar{\Upsilon }+\left(B_{\epsilon }-{\hat{B}}_{w}\right)Y_{l} \end{array}\right],$$


$${\bar{\Psi }}_{62}=\left[\begin{array}{c} -B_{\epsilon }Y_{l}\\ {\hat{A}}_{\epsilon }\bar{\Upsilon }-\left(B_{\epsilon }-{\hat{B}}_{w}\right)Y_{l} \end{array}\right],{\bar{\Psi }}_{63}=\left[\begin{array}{c} -B_{\epsilon }Y_{l}\\ -\left(B_{\epsilon }-{\hat{B}}_{\epsilon }\right)Y_{l}\; \end{array}\right],{\bar{\Psi }}_{67}=\left[\begin{array}{cc} -\bar{\Upsilon } & *\\ {}* & -\bar{\Upsilon } \end{array}\right],$$


$${\bar{\Psi }}_{23}=-\upsilon_{1}{\bar{\Upsilon }}^{-1}I{\bar{\Upsilon }}^{-1},{\bar{\Psi }}_{34}=-\upsilon_{1}{\bar{\Upsilon }}^{-1}I{\bar{\Upsilon }}^{-1},$$


$${\bar{\Psi }}_{45}=\upsilon_{2}{\bar{\Upsilon }}^{-1}I{\bar{\Upsilon }}^{-1},{\bar{\Psi }}_{56}=\upsilon_{2}{\bar{\Upsilon }}^{-1}I{\bar{\Upsilon }}^{-1},K_{l}=Y_{l}{\bar{\Upsilon }}^{-1}.$$



**Proof.** Define $P=diag\{\bar{P},\bar{P}\}$, $\bar{\Upsilon }={\bar{P}}^{-1}$, $\Upsilon =diag\{\bar{\Upsilon },\bar{\Upsilon }\}$, $D=diag\{\Upsilon ,\bar{\Upsilon },\bar{\Upsilon },\bar{\Upsilon },\bar{\Upsilon },\Upsilon \}$. Just left and right multiply D on (Equation 17)–(Equation 19), then (Equation 30)–(Equation 32) can be obtained. □

## 4. Numerical Examples

In this part, a numerical simulation is used to prove the effectiveness of the designed control scheme for the networked interval type-2 fuzzy system. A nonlinear mass-spring system is given as:$$\begin{array}{cc} \; & {\dot{\varrho }}_{1}=\varrho_{2},\\ \; & \varrho_{2}=-0.01\varrho_{1}-0.67\varrho_{1}^{3}+u, \end{array}$$
where $\varrho_{1}\in \left[-1,1\right]$. If the nonlinear mass-spring system is discretized with sampling period h=0.1, then the discrete fuzzy system is:(33)$$\begin{array}{c} x\left(t+1\right)=\sum_{\epsilon =1}^{2}h_{\epsilon }\left(f\left(x\right)\right)\left[A_{i}x\left(t\right)+B_{i}x\left(t\right)\right], \end{array}$$
where:$$A_{1}=\left[\begin{array}{cc} 1.0000 & 0.1000\\ -0.0010 & 1.0000 \end{array}\right],\;B_{1}=\left[\begin{array}{c} 0.0050\\ 0.100 \end{array}\right],$$
$$A_{2}=\left[\begin{array}{cc} 0.9966 & 0.0999\\ -0.0679 & 0.9966 \end{array}\right],\;B_{2}=\left[\begin{array}{c} 0.0050\\ 0.0999 \end{array}\right],$$
$$h_{\epsilon }\left(x\left(k\right)\right)={\underline{\kappa }}_{i}{\underline{h}}_{\epsilon }\left(x\left(k\right)\right)+{\bar{\kappa }}_{\epsilon }{\bar{h}}_{\epsilon }\left(x\left(k\right)\right),\;h_{2}=1-h_{1},$$
$${\bar{h}}_{1}\left(f\left(x\right)\right)=\frac{1}{1+exp(-\varphi_{2}x_{1}\left(t\right))},\;{\underline{h}}_{2}\left(f\left(x\right)\right)=1-{\bar{h}}_{1}\left(f\left(x\right)\right),$$
$${\underline{h}}_{1}\left(f\left(x\right)\right)=\frac{1}{1+exp(-\varphi_{1}x_{1}\left(t\right))},\;{\bar{h}}_{2}\left(f\left(x\right)\right)=1-{\underline{h}}_{1}\left(f\left(x\right)\right),$$
and the membership functions of the controller is:$$\eta_{l}\left(g\left(\hat{x}\right)\right)={\underline{\kappa }}_{l}{\underline{\eta }}_{l}\left(x\left(k\right)\right)+{\bar{\kappa }}_{l}{\bar{\eta }}_{l}\left(x\left(k\right)\right),\;\eta_{2}=1-\eta_{1},$$
$${\bar{\eta }}_{1}\left(g\left(\hat{(}x\right)\right))=0.98exp\left(-\varphi_{1}x_{1}\left(t\right)\right),\;{\underline{\eta }}_{2}\left(g\left(\hat{(}x\right)\right))=1-{\bar{\eta }}_{1}\left(g\left(\hat{(}x\right)\right)),$$
$${\underline{\eta }}_{1}\left(g\left(\hat{(}x\right)\right))=0.98exp\left(-\varphi_{2}x_{1}\left(t\right)\right),\;{\bar{\eta }}_{2}\left(g\left(\hat{(}x\right)\right))=1-{\underline{\eta }}_{1}\left(g\left(\hat{(}x\right)\right)),$$
with $\varphi_{1}=1,\varphi_{2}=2.$

**Case 1:** Assume that the constant matrices of the predictor (Equation 15) and the system (Equation 14) are the same, that is $\hat{A}=A,\;\hat{B}=B$. Given parameter $\gamma_{1}=0.8$ and $\gamma_{2}=0.95$ and ensure that $\eta_{l}\left(\hat{x}\right)-\gamma_{l}h_{l}\left(x\right)\geq 0$. Set the event-triggered scalars as ρ=4,δ=0.8,τ=0.3. By using LMI, controller gains can be obtained as:$$K_{1}=\left[\begin{array}{cc} -0.1581 & -0.0851 \end{array}\right],\;K_{2}=\left[\begin{array}{cc} -0.1481 & -0.0441 \end{array}\right].$$

Suppose that the initial state is $x_{0}={\left[0.5,-0.5\right]}^{T}$, the sampling period is 0.1 s, and the simulation time is 100 s. Packet loss occurs randomly in the communication network, and the maximum number of consecutive packet loss is σ=10. Figure 2, Figure 3, Figure 4 and Figure 5 show the evolution of the system state, event-triggered intervals, the evolution of the variable ϕ(k) and the data dropout instants in case 1, respectively.

**Case 2:** Assume that the constant matrices of the predictor (Equation 15) and the system (Equation 14) are different, that is $\hat{A}=1.02*A,\;\hat{B}=0.9*B$. Given parameter $\gamma_{1}=0.8$ and $\gamma_{2}=0.95$ and ensure that $\eta_{l}\left(\hat{x}\right)-\gamma_{l}h_{l}\left(x\right)\geq 0$. Set the event-triggered scalars as ρ=4,δ=0.8,τ=0.3. By using LMI, controller gains can be obtained as:$$K_{1}=\left[\begin{array}{cc} -0.3145 & -0.8307 \end{array}\right],\;K_{2}=\left[\begin{array}{cc} -0.0152 & -0.4473 \end{array}\right].$$

Suppose that the initial state is $x_{0}={\left[0.5,-0.5\right]}^{T}$, the sampling period is 0.1s, and the simulation time is 100s. Packet loss occurs randomly in the communication network, and the maximum number of consecutive packet loss is σ=10. Figure 6, Figure 7, Figure 8 and Figure 9 show the evolution of the system state, event-triggered intervals, the evolution of the variable ϕ(k) and the data dropout instants in case 2, respectively.

Because the packet loss occurs randomly, the ETM of the two cases can not be compared. Therefore, we set the packet loss moments of the two cases to be the same, and Table 1 shows the frequency of event-triggered dynamic ETM1 (Equation 3) and static ETM in case 1 and case 2. It can be seen that dynamic ETM1 (Equation 3) has a lower event-triggered frequency than static ETM.

## 5. Conclusions

An IT2 T-S fuzzy model is used for modeling a class of NCSs, and an FETPC design method for systems considered with imperfect premise matching is proposed. The dynamic ETM1 has been used to reduce the network load and maintain certain control performance. The designed FETPC can predict the state of the system between two successful transmissions. By choosing the Lyapunov function and some inequalities, sufficient conditions have been obtained to ensure the property of the closed-loop system, and a clear representation of the event-triggered predictive controller is presented. Finally, numerical simulations are used to illustrate the effectiveness of the designed method. In this paper, network delay is not considered, and the stability of NCSs with network-induced delay will be studied in future work. There are more practical factors that we need to consider, such as the uncertainty of system parameters, the failure of physical devices, and the quantification of network signals. All these will inspire our future work.

## Figures and Tables

**Figure 1 entropy-23-01452-f001:**
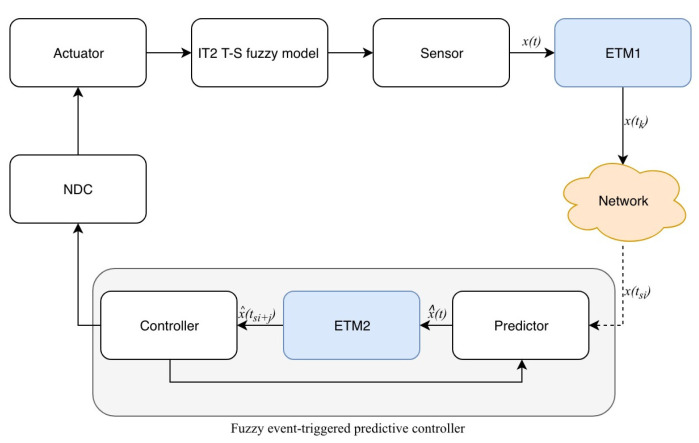
Diagram of the considered IT2 fuzzy systems.

**Figure 2 entropy-23-01452-f002:**
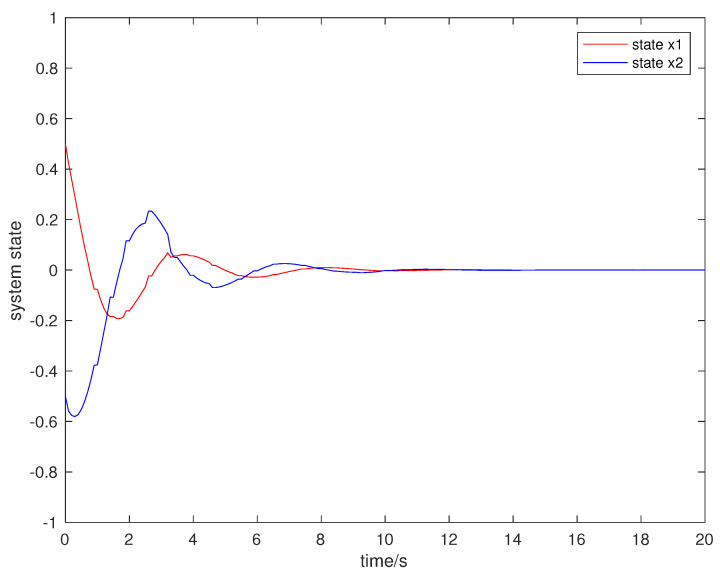
System state in case 1.

**Figure 3 entropy-23-01452-f003:**
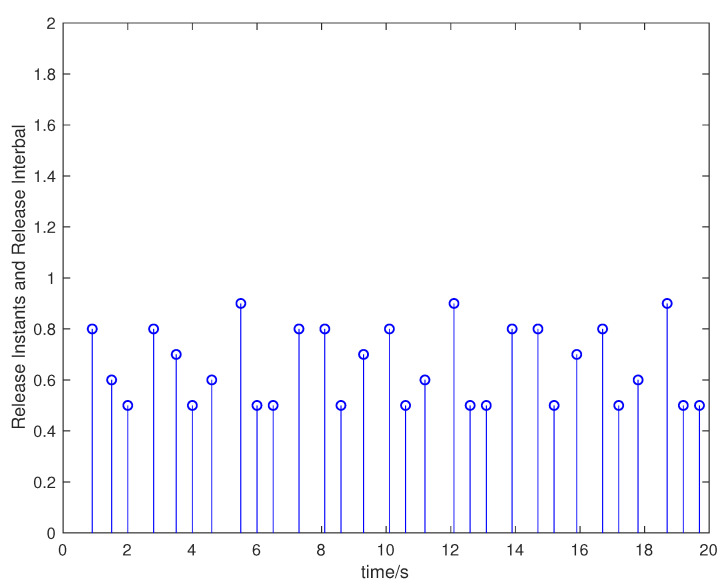
Event-triggered intervals in case 1.

**Figure 4 entropy-23-01452-f004:**
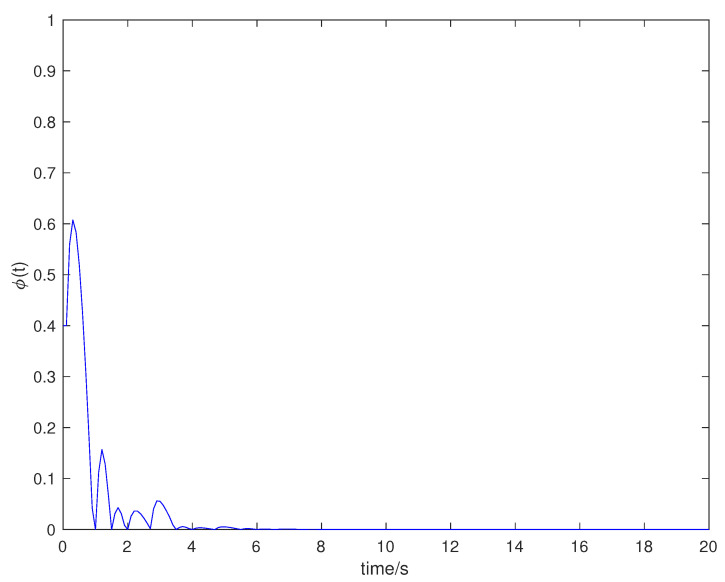
Dynamic variable ϕ(k) in case 1.

**Figure 5 entropy-23-01452-f005:**
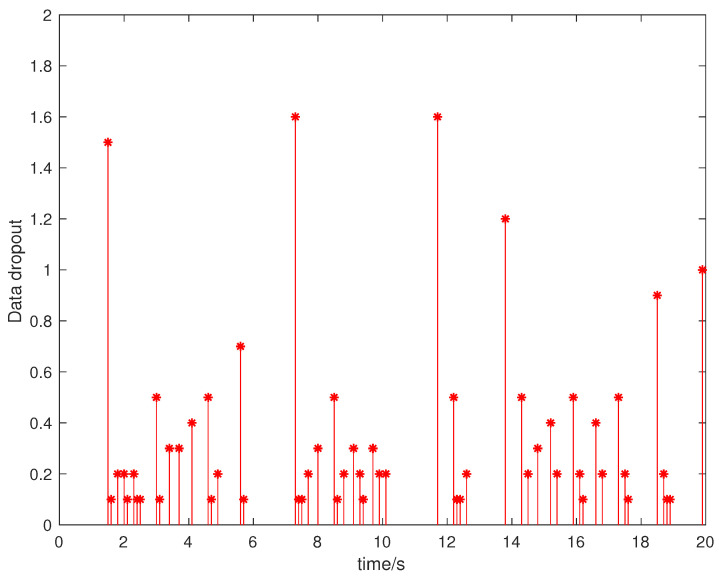
Data dropout instants and intervals in case 1.

**Figure 6 entropy-23-01452-f006:**
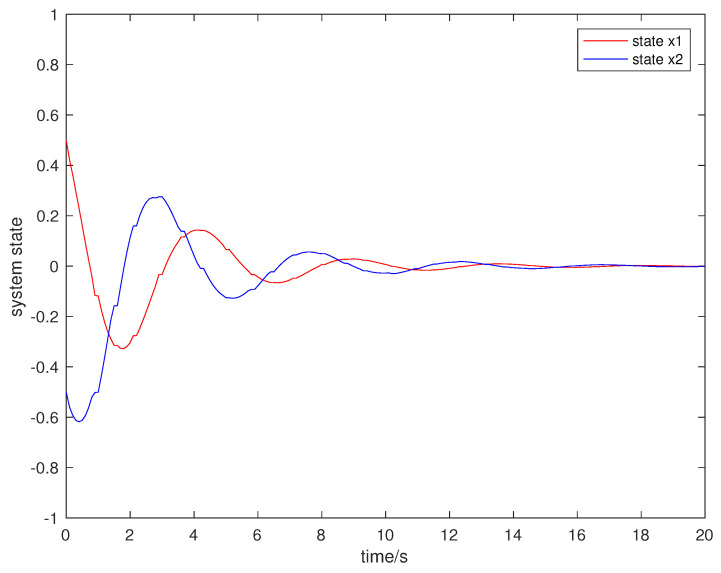
System state in case 2.

**Figure 7 entropy-23-01452-f007:**
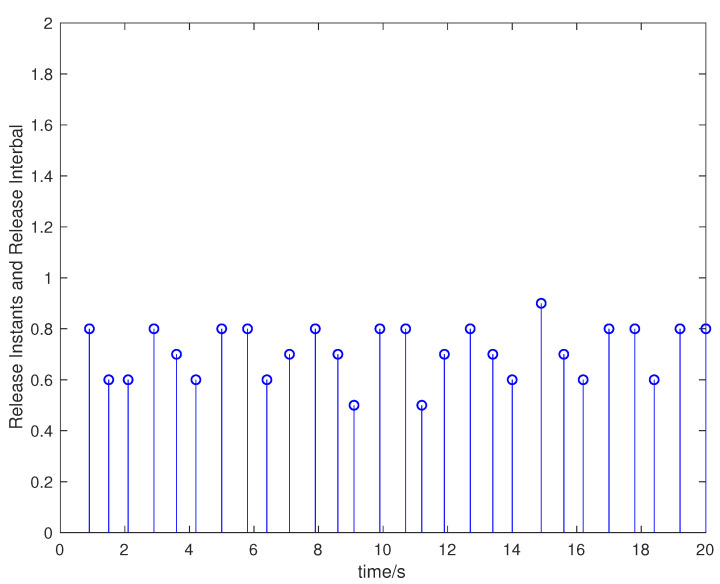
Event-triggered intervals in case 2.

**Figure 8 entropy-23-01452-f008:**
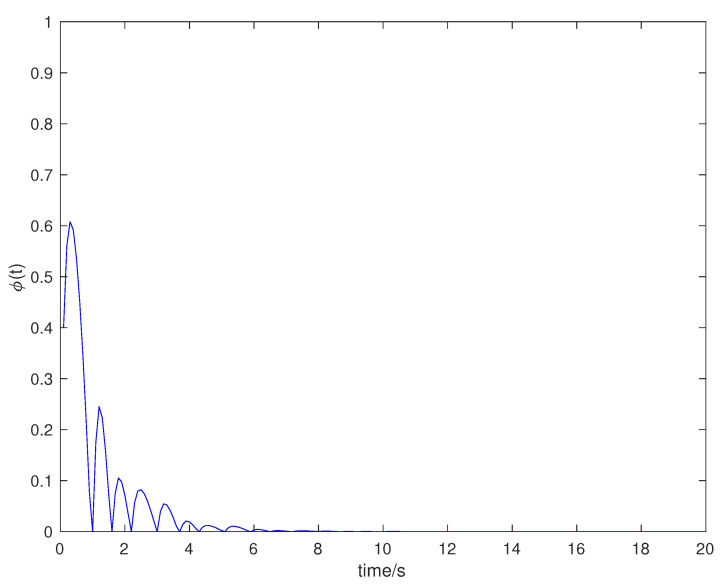
Dynamic variable ϕ(k) in case 2.

**Figure 9 entropy-23-01452-f009:**
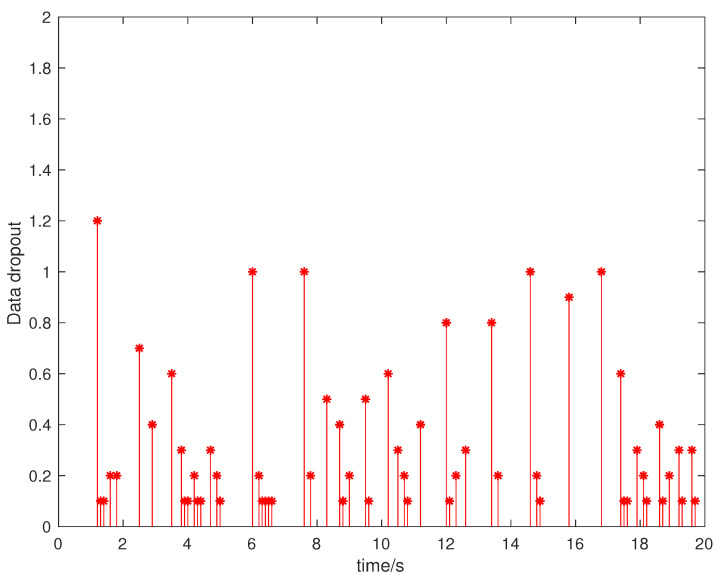
Data dropout instants and intervals in case 2.

**Table 1 entropy-23-01452-t001:** The frequency of event-triggered mechanism.

ETM	Frequency	The Sampling Period
dynamic ETM (ϕ(k)≠0) in case 1	27	200
static ETM (ϕ(k)=0) in case 1	52	200
dynamic ETM (ϕ(k)≠0) in case 2	30	200
static ETM (ϕ(k)=0) in case 2	56	200

## Data Availability

Not applicable.

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
