# Peer review of "Dynamic Event-Triggered Predictive Control for Interval Type-2 Fuzzy Systems with Imperfect Premise Matching"

_entropy, 2021, doi:10.3390/e23111452_

Round 1

Reviewer 1 Report

The paper presents a solid work about the use of a Type-II fuzzy system for dynamic event-triggered network.

Some comments are:

- The authors should analyse and discuss other applications of type-II fuzzy systems in sensor networks.

- The authors should discuss how the communication affects their proposal.

Author Response

Comment: The paper presents a solid work about the use of a Type-II fuzzy system for dynamic event-triggered network.

Reply: Thank you very much for your careful reading and the kind encouragement of our work.

Comment:  The authors should analyse and discuss other applications of type-II fuzzy systems in sensor networks.

 Reply: Thank you for your valuable comment. In the revision, we have added the application of the fuzzy model. The T-S fuzzy model can be used to represent some systems with nonlinear dynamics by the local linear subsystems under several IF-Then rules. In addition, it can also solve some nonlinear problems. For example, nonlinear disturbances can be represented by the local linear disturbances. Therefore, the fuzzy model is widely used in practice, and it is very meaningful to study it. The application of type-II fuzzy systems in sensor networks will be one of future directions that deserve special attentions.

Commend: The authors should discuss how the communication affects their proposal.

  Reply:  Thank you for your suggestions. The communication network designed in this study exists between the sensor and the controller, which only affects the signal transmission between them, but not other components. The data sampled by the sensor is transmitted to the controller under the ETM1 through the communication network. Moreover, there is packet loss in the communication network, which will affect the transmission of data. In this paper, the predictor is introduced to effectively compensate for the impact of network packet loss. Pertinent other consideration (e.g. cyber attacks) about communication will be paied more attention in the future.

Reviewer 2 Report

The topic is interesting, but I would suggest the following comments for improvements:

- Please give a frank account of the strengths and weaknesses of the proposed research method.

- The research motivations are unclear and rather vague. The motivation, what the authors make and how they make it, is very confusing and arguable. The authors must also clearly discuss the significance of the research problem in the first section.

- The authors need to fully discuss insightful and practical implications of their approach.

- The authors need to include a good literature survey to show exactly what is novel about their study. In fact, I would like a clear discussion on the current literature versus the unique contribution of the paper.

Round 2

Reviewer 2 Report

My comments have been addressed. So, the paper can be accepted.